# Are Dried and Powdered *Moringa oleifera* Lam. Leaves Susceptible to Moths That Feed on Stored Products?

**DOI:** 10.3390/insects12070610

**Published:** 2021-07-05

**Authors:** Lidia Limonta, Daria Patrizia Locatelli

**Affiliations:** Department of Food, Environmental and Nutritional Sciences, Università degli Studi di Milano, 20133 Milano, Italy; daria.locatelli@unimi.it

**Keywords:** development, *Moringa oleifera*, larvae, Rustywave moth, Pyralid moths

## Abstract

**Simple Summary:**

In recent years, *Moringa oleifera* leaves have been increasingly introduced into the diets of human populations that are affected by malnutrition and as a dietary supplement in Western countries. The leaves can be stocked in storage spaces that contain other herbs or food that can be infested by Pyralid moths, such as the polyphagous almond moth, rice moth, and Indian meal moth, as well as by the Geometrid Rustywave moth, which thrives on dried herbs. This paper describes laboratory tests of the susceptibility of dried and powdered leaves of *M. oleifera* to moths that feed on stored products. Eggs of the different species were added to dried *M. oleifera* leaves, powdered leaves, and an artificial diet; dried or powdered leaves were added to the latter to understand the effects on the development of the moths. The tests were carried out under the optimal temperature and relative humidity conditions for these species. The numbers of adults that emerged and the development periods were recorded. The results showed that powdered Moringa leaves were not susceptible to moth attacks, whereas dried leaves were damaged only by Rustywave moths. The explanation for why *M. oleifera* leaves are not susceptible to Pyralid moths and for why few Rustywave moths can complete the development from egg to adult is attributable to both nutritional deficiency and to secondary metabolites.

**Abstract:**

The leaves of *Moringa oleifera* are increasingly used as a food supplement in several countries due to their nutritional composition, which is rich in protein, vitamins, and mineral salts. Foodstuffs can be damaged by several pests when stored in environments with temperatures that are favorable to insect development; therefore, the susceptibility of *M. oleifera* leaves to attacks of moths that feed on stored products was tested. Tests were carried out on Pyralid *Cadra cautella*, *Corcyra cephalonica*, and *Plodia interpunctella*, as well as Geometrid *Idaea inquinata*, which were reared on dried whole or powdered *M. oleifera* leaves, an artificial diet, or an artificial diet supplemented with dried or powdered leaves. The numbers of adults and the development periods with the different diets were recorded. *M. oleifera* leaves were unsuitable as a rearing medium for all of the species except *I. inquinata*, although only a few individuals of this species reached the adult stage. The use of an artificial diet of which one-quarter consisted of dried and powdered leaves did not affect the number of progeny or on the biological cycle, showing that the effect was due to the nutritional composition, as well as to the toxic effect. The storage of *M. oleifera* as powdered leaves is recommended in order to preserve the nutritional characteristics and avoid damage caused by moth larvae.

## 1. Introduction

*Moringa oleifera* Lam. is a tree that is native to South Asia and is widely grown in tropical areas. Local populations have used all of the different parts of the tree for thousands of years. The roots and flowers are used in traditional medicine, the seeds are employed for water purification, and the trees are planted to avoid soil erosion.

The leaves of *M. oleifera* have been introduced into the diets of several human populations to address malnutrition problems, because they contain good amounts of proteins, calcium, iron, potassium, vitamins, and β-carotene. The leaves present a very high content of fiber, antioxidant and bioactive compounds, flavonoids, and phenolic acids [1,2,3]; therefore, they are used as a food supplement in Western countries. The leaves are dried before storage and marketed whole or powdered. *M. oleifera* is mainly grown in South Asia, but it is also cultivated in Africa, South America, and the Caribbean. In these countries, vegetables are usually naturally air-dried in processing plants that are not completely isolated from the external environment; insects that feed on stored products can easily invade in such cases, causing huge losses by reducing the economic value and compromising the use for human nutrition. Moths that feed on stored products have already been reported to cause damage to dried herbs [4,5,6]; therefore, we chose to investigate three polyphagous moths—*Cadra cautella* (Walker), *Corcyra cephalonica* (Stainton), and *Plodia interpunctella* (Hübner) (Pyralidae)—and one species that feeds mainly on dried herbs—*Idaea inquinata* (Scopoli) (Geometridae). These Pyralid species can develop in different stored foods [7], such as cereal grains [8], flour [9], and dried fruit [10,11,12]. The damage caused by Pyralid moths is not only economically important, but it also compromises food safety, because it was demonstrated that these species produce allergens that affect human health [13,14]. The Geometrid *I. inquinata* is mostly found in hay lofts and in barns [15,16,17], but more recent research has shown that this species is considered as a potential pest for cereals and some of their derivatives, as well as for medicinal plants [18].

In this research, the susceptibility of dried and powdered leaves of *M. oleifera* to some species of moths was verified. Tests were also carried out by using an artificial diet, of which one-quarter consisted of dried and powdered leaves when rearing in the laboratory in order to determine if the effects of Moringa leaves on the development of these species were linked to their nutritional composition or to the presence of secondary components.

## 2. Materials and Methods

### 2.1. Insect Rearing

Stock rearing of *Cadra cautella, Corcyra cephalonica, Plodia interpunctella,* and *Idaea inquinata* has been maintained for 10 years at DeFENS, Università degli Studi di Milano “La Statale”. The insects were reared in Petri dishes (Ø 15 cm), placed in a growth chamber (Piardi mod. CFT600) at 26 ± 1 °C with 60 ± 5% RH, and a photoperiod of 16:8 (light–dark).

*C. cautella*, *C. cephalonica,* and *P. interpunctella* were reared on an artificial diet consisting of glycerol, honey, cornmeal, bran, wheat meal, wheat germ, and yeast [19]. The artificial diet of *I. inquinata* comprised the same ingredients, but with a higher bran percentage [20]. Proximate analyses were performed on 50 g of the artificial diets of *I. inquinata* and Pyralidae to determine their nutritional value (two replicates, expressed as means ± S.D.). Different methods were used: the fiber content was analyzed according to the method of Prosky et al. [21]; carbohydrates were determined with the method of Rocklin and Pohl [22]; and the methods of the Association of Analytical Communities and the American Association for Clinical Chemistry were performed to measure proteins [23], fats [24], and ashes [25].

### 2.2. Egg Collection

For each species, newly emerged adults were placed in a glass jar (1.7 L), which was closed with tulle that was fixed with a plastic band, and the jar was turned upside down and placed on an open Petri dish. The bottom was covered with black paper. After 24 h, the eggs were collected for the tests.

### 2.3. Tests

For each species—namely, *Cadra cautella*, *Corcyra cephalonica*, *Plodia interpunctella,* and *Idaea inquinata*—one hundred eggs were reared on 10 g of the following six media: artificial diet, dried *Moringa oleifera* leaves, powdered *M. oleifera* leaves, ½ dried *M. oleifera* leaves + ½ powdered leaves, ¼ dried *M. oleifera* leaves + ¾ artificial diet, ¼ powdered *M. oleifera* leaves + ¾ artificial diet. *M. oleifera* was cultivated and dried in Haiti; dried leaves were powdered in our laboratory.

The rearing media were placed in PVC containers (height: 5 cm, diameter: 7 cm). A hole that was 2 cm in diameter was added to the lids of the containers and closed with a wire net to allow gas exchange. Before the tests, the PVC containers with the six media plates were maintained in the testing conditions for a week. One hundred eggs that were laid in 24 h were added. For each species and each rearing medium, five replicates were carried out. The testing conditions were kept at 26 ± 1 °C with 60 ± 5% RH and a photoperiod of 16:8 (light–dark). After 20 days, the replicates were checked daily. The adults that emerged were counted and removed, and the development time was recorded.

To assess the normality distribution of data, a Kolmogorov–Smirnov test was carried out, and the normally distributed data were subjected to one-way ANOVA and the least significant difference (LSD) test (IBM SPSS Statistics 26).

## 3. Results

Adults of the *Cadra cautella*, *Corcyra cephalonica*, and *Plodia interpunctella* groups were not observed in the tests on the dried or powdered *Moringa oleifera* leaves (Table 1). Only a few individuals of *Idaea inquinata* developed on dried leaves of *M. oleifera*; none developed on the powdered leaves. All the test data were normally distributed.

The number of adults of *C. cautella* (Table 1, Figure 1) reared on ¼ dried or powdered leaves and ¾ of the artificial diet was significantly lower than the number of adults reared on the artificial diet, but the development time was not influenced (one-way ANOVA: Adults F _2, 12_ = 155.730, *p* < 0.05; time F _2, 12_ = 1.334, P = 0.3 n.s.). The number of adults and the development time of *C. cephalonica* (Table 1, Figure 2) were not influenced by the artificial diet with one-quarter of dried or powdered leaves (one-way ANOVA: adults F _2, 12_ = 0.947, P = 0.415 n.s.; time F _2, 12_ = 3.295, P = 0.072 n.s.). In the tests on *P. interpunctella* (Table 1, Figure 3), the addition of powdered leaves to the artificial diet at a proportion of one-quarter did not influence the number of adults, but caused a significant increase in the development time, whereas the addition of dried leaves resulted in fewer adults emerging and a shorter development period (one-way ANOVA: adults F _2, 12_ = 2.458, P = 0.127 n.s.; time F _2, 12_ = 5.203, *p* < 0.05).

The numbers of adults and the development times of *I. inquinata* (Table 1, Figure 4) reared on the artificial diet alone and with the addition of dried leaves were not significantly different (one-way ANOVA: Adults F _4, 20_ = 2346.23, *p* < 0.05; time F _4, 20_ = 11.695, *p* < 0.05). The addition of powdered leaves to the artificial diet caused a decrease in the number of adults, but not in the development time. Fewer adults and a longer development time were observed when using dried leaves alone and when using the mixture with powdered leaves of *M. oleifera*.

The main nutrients of the artificial diets are summarized in Table 2. In the case of the main nutrients of *M. oleifera*, we refer to Leone et al. [26], considering the values of Moringa from Haiti. The highest protein and lipid contents were recorded in *M. oleifera* (20.8% and 7.0%, respectively), and the lowest were recorded in the artificial Pyralid diet (8.6% and 2.0%, respectively). The starch content was higher in the artificial diets of Pyralid and *I. inquinata* (29.3% and 22.0%, respectively), and it was lower in the *M. oleifera* leaves (13.7%). The total fiber was high in *M. oleifera* (37.6%) and in the artificial *I. inquinata* diet (27.6); it was low (9.7%) in the artificial Pyralid diet. The moisture content was 12.4 (±0.09) in the Pyralid diet, 12.8 (±0.01) in the *I. inquinata* diet, and 8.8 (±0.1) in dried Moringa leaves.

## 4. Discussion

Dried and powdered *Moringa oleifera* leaves were unsuitable as rearing media for *Cadra cautella*, *Corcyra cephalonica*, and *Plodia interpunctella*. Only a few *Idaea inquinata* specimens developed on the dried leaves. *M. oleifera* leaves lack essential nutrients for the postembryonic development of the Pyralid moths. When comparing artificial diets and the *M. oleifera* leaves, it should be noted that they have different nutritional compositions, because *M. oleifera* leaves have higher contents of protein, lipids, and ash, and lower starch and moisture contents compared to the other diets. The artificial diets also contained glycerol, which contributes some nutritive factors, increases the diet water content, and is considered a booster for *P. interpunctella* larval growth [27].

Powdered *M. oleifera* leaves were unsuitable as a rearing medium for *I. inquinata,* as has also been observed for different flours [18]; in this case, the cause was the fine particle size, rather than the nutritional composition.

The effect on the moths’ development can be ascribed to the nutritional composition, although it must be considered that the powdered leaves of *M. oleifera* exert insecticidal and repellent activities on the larvae and adults of *Trogoderma granarium* [28,29]. The leaves contain catechol, tannins, gallic tannins, steroids, triterpenoids, flavonoids, saponins, anthraquinones, and alkaloids [30], and secondary metabolite, such as glucosinolates and isothiocyanates. It has been demonstrated that tannins and saponins exert toxic action on insects [31,32,33,34,35]. In our experiments, the numbers of adults and the development periods of *Corcyra cephalonica* and *Idaea inquinata* were not affected by the addition of *M. oleifera* leaves to the artificial diet in a proportion of one-quarter; therefore, the number of leaves was insufficient for the secondary metabolites to exert their toxic action.

Concerning the behavior of the larvae on the different diets, we observed that the first instar larvae of *C. cautella*, *C. cephalonica,* and *P. interpunctella* reared on dried or powdered moringa leaves wandered until they starved. In a double-choice test with 50 eggs in the center of a Petri dish—dried *M. oleifera* leaves on one side and the artificial diet on the other side—we also observed that newly hatched first instar larvae were attracted by and fed only on the artificial diet (authors’ observations).

*I. inquinata* feed on some dried herbs [18]; when the dried *M. oleifera* leaves were the rearing medium, 15% of the *I. inquinata* eggs developed into adults. The secondary metabolites of *M. oleifera* likely interfered with larval development, as was observed in *Camellia sinensis* and *Vitis vinifera*, which are characterized by their high tannin contents and were unsuitable as rearing media for *I. inquinata*. Tannins act on ß-glucosidase and esterase [35], act as an antifeedant in *Leptinotarsa decemlineata* [36], and cause a delay in larval development when added to the diet of Lepidoptera larvae [37].

Dried herbs in warehouses can be infested by several pests [38,39,40], including *C. cautella* and *P. interpunctella*; these species were not able to develop on dried and powdered *M. oleifera* leaves. *M. oleifera* leaves can be safely stored as far as Pyralid moths are concerned, but the dried leaves can be infested by *I. inquinata*. This species can compromise the use of the leaves with its exuviae and excrements, especially if the Moringa leaves are stored with other herbs that can be infested by this species. The results obtained in this research show that in order to preserve the nutritional characteristics and avoid damage, the storage of powdered leaves is recommended. Although *M. oleifera* leaves are hardly susceptible to pest infestation, it is recommended to carefully check dried herbs, monitor warehouses carefully, avoid debris accumulation, and protect windows with wire-nets.

## Figures and Tables

**Figure 1 insects-12-00610-f001:**
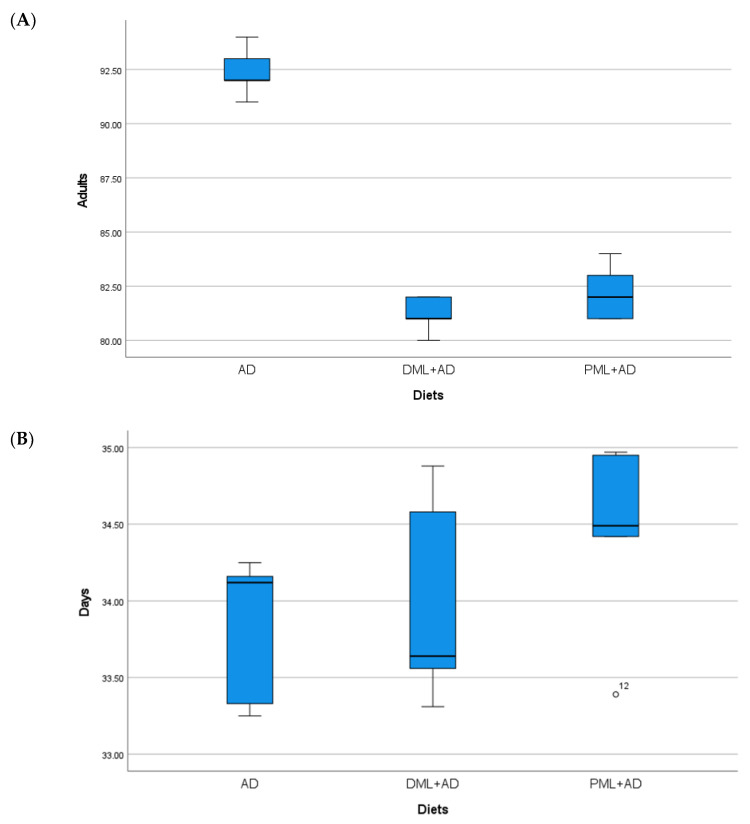
(**A**) Mean number (±S.E.) of adults; (**B**), mean development times (days ± S.E.) of *Cadra cautella* (Walker), from 100 eggs reared on artificial diet (AD), ¼ dried *Moringa oleifera* leaves + ¾ artificial diet, ¼ (DML+AD), powdered *M. oleifera* leaves + ¾ artificial diet (PML+AD), at 26 ± 1 °C with 60 ± 5% RH and a photoperiod of 16:8 (light–dark).

**Figure 2 insects-12-00610-f002:**
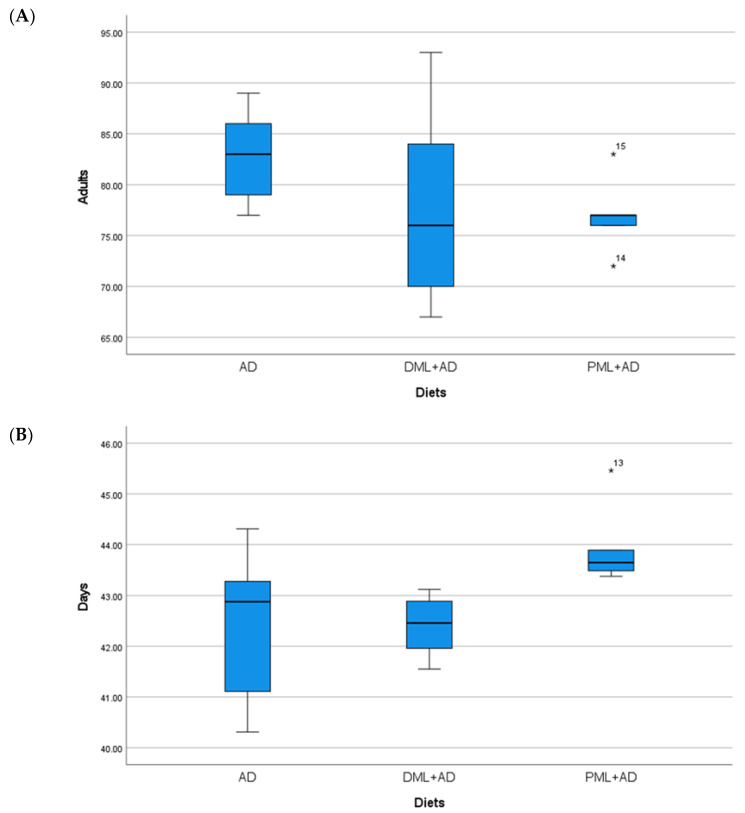
(**A**) Mean number (±S.E.) of adults; (**B**) mean development times (days ± S.E.) of *Corcyra cephalonica* (Stainton), from 100 eggs reared on artificial diet (AD), ¼ dried *Moringa oleifera* leaves + ¾ artificial diet, ¼ (DML+AD), powdered *M. oleifera* leaves + ¾ artificial diet (PML+AD), at 26 ± 1 °C with 60 ± 5% RH and a photoperiod of 16:8 (light–dark).

**Figure 3 insects-12-00610-f003:**
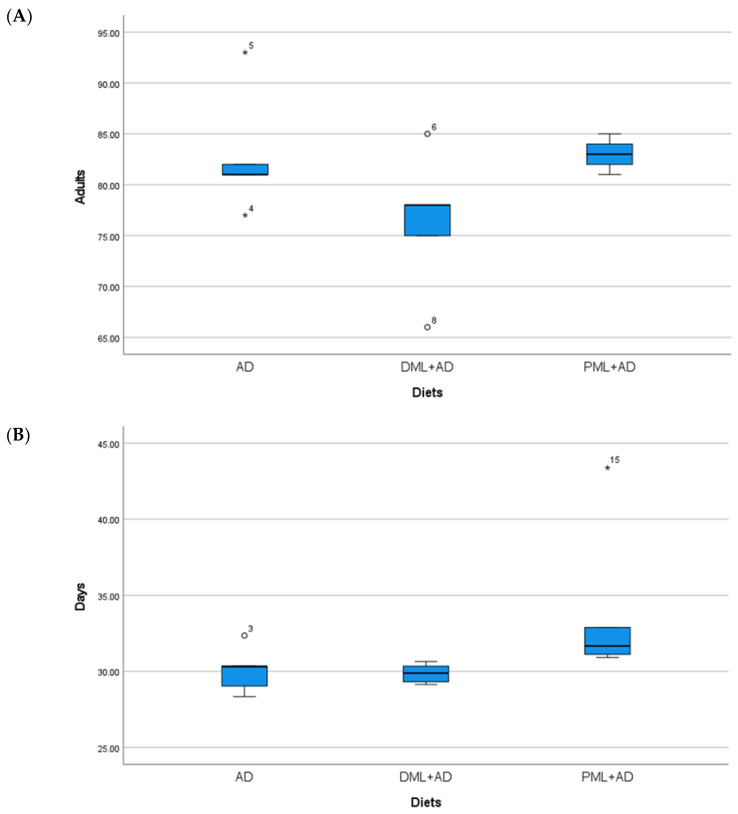
(**A**) Mean number (±S.E.) of adults; (**B**) mean development times (days ± S.E.) of *Plodia interpunctella* (Hübner), from 100 eggs reared on artificial diet (AD), ¼ dried *Moringa oleifera* leaves + ¾ artificial diet, ¼ (DML+AD), powdered *M. oleifera* leaves + ¾ artificial diet (PML+AD), at 26 ± 1 °C with 60 ± 5% RH and a photoperiod of 16:8 (light–dark).

**Figure 4 insects-12-00610-f004:**
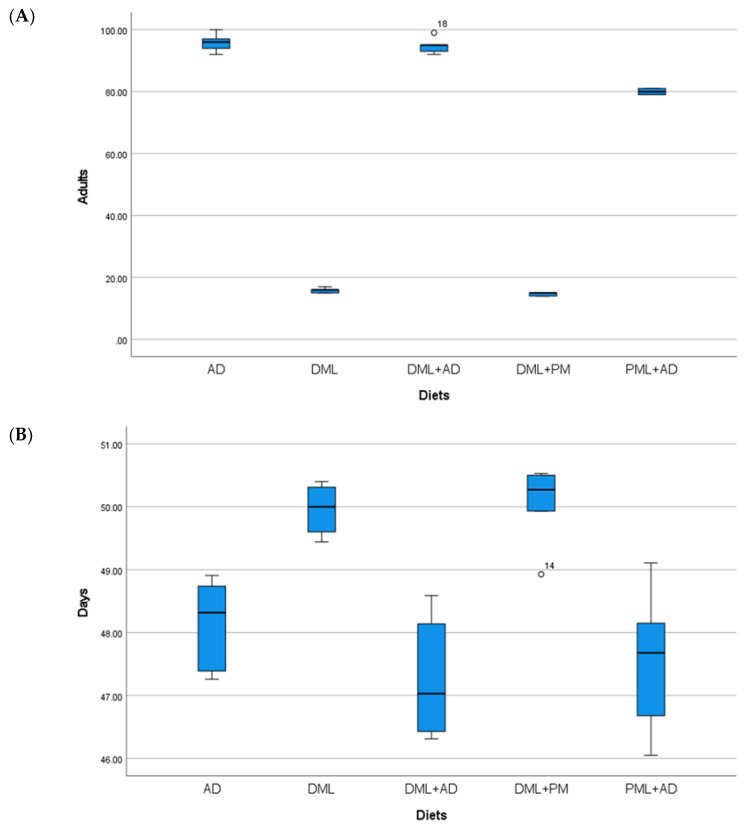
(**A**) Mean number (±S.E.) of adults; (**B**) mean development times (days ± S.E.) of *Idaea inquinata* (Scopoli), from 100 eggs reared on artificial diet (AD), dried *Moringa oleifera* leaves (DML), powdered *M. oleifera* leaves (PML), ½ dried *M. oleifera* leaves + ½ powdered leaves (DML+PM), ¼ dried *M. oleifera* leaves + ¾ artificial diet (DML+AD), ¼ powdered *M. oleifera* leaves + ¾ artificial diet (PML+AD), at 26 ± 1 °C with 60 ± 5% RH and a photoperiod of 16:8 (light–dark).

**Table 1 insects-12-00610-t001:** Mean numbers (±S.E.) of adults and the development times (days ± S.E.) of *Cadra cautella* (Walker), *Corcyra cephalonica* (Stainton), *Plodia interpunctella* (Hübner), and *Idaea inquinata* (Scopoli), from 100 eggs reared on the artificial diet (AD), dried *Moringa oleifera* leaves (DML), and powdered *Moringa oleifera* leaves (PML) at 26 ± 1 °C with 60 ± 5% RH and a photoperiod of 16:8 (light–dark). Means followed by different letters are significantly different (LSD test).

Diets.	*Cadra cautella*		*Corcyra cephalonica*		*Plodia interpunctella*		*Idaea inquinata*	
	Adults	Days	Adults	Days	Adults	Days	Adults	Days
AD	92.4 ± 0.51 a	33.8 ± 0.22	82.8 ± 2.20	42.4 ± 0.73	82.8 ± 2.69 a	30.1 ± 0.69 a,b	95.8 ± 1.36 a	48.1 ± 0.34 b
DML	0.0 ± 0.00	-	0.0 ± 0.00	-	0.0 ± 0.00	-	15.8 ± 0.37 c	49.9 ± 0.18 a
PML	0.0 ± 0.00	-	0.0 ± 0.00	-	0.0 ± 0.00	-	0.0 ± 0.00	-
½DML + ½PML	0.0 ± 0.00	-	0.0 ± 0.00	-	0.0 ± 0.00	-	14.6 ± 0.24 c	50.0 ± 0.29 a
¼DML + ¾AD	81.2 ± 0.37 b	34.2 ± 0.30	78.0 ± 4.74	42.4 ± 0.29	76.4 ± 3.07 b	29.9 ± 0.29 b	94.8 ± 1.20 a	47.3 ± 0.47 b
¼PML + ¾AD	82.2 ± 0.58 b	34.4 ± 0.29	77.0 ± 1.76	44.0 ± 0.38	83.0 ± 0.71 a	32.0 ± 0.50 a	80.0 ± 0.45 b	47.5 ± 0.54 b

**Table 2 insects-12-00610-t002:** Nutritional characterization of the artificial diets of *Idaea inquinata* and of the Pyralid moths (mean ± S.D.).

Nutrients	Idaea	Pyralidae
Proteins	13.9 ± 0.261	8.6 ± 0.05
Lipids	2.9 ± 0.12	2.0 ± 0.19
Starch	22.0 ± 0.79	29.3 ± 1.13
Soluble carbohydrates *	8.1 ± 0.27	13.8 ± 0.21
Total fiber	27.6 ± 1.01	9.7 ± 0.37
Ashes	3.6 ± 0.08	1.4 ± 0.02

* Sum of glucose, fructose, and sucrose.

## Data Availability

Data are available on request.

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
