# Peer review of "Are Dried and Powdered Moringa oleifera Lam. Leaves Susceptible to Moths That Feed on Stored Products?"

_insects, 2021, doi:10.3390/insects12070610_

Round 1

Reviewer 1 Report

The author's work titled “Are dried and powdered Moringa oleifera Lam. leaves susceptible to moths that feed on stored products?” describes the survivability of four stored product pests on the leaves of Moringa oleifera, an artificial diet and a meridic diet containing M. oleifera leaves. The author's methodology shows credibility that this method as well as assays developed are suitable to investigate the author's research questions. The data presented are sound and does justify the author's conclusion, supporting the claim that M. oleifera leaves can be stored and safe from Pyralid moth infestation, however the dried leaves can be infested by I. inquinata. The analysis and interpretation of the author's finding are simplified and comprehensible. Overall, the manuscript was well prepared and this study can be replicated with the information provided. Therefore, this work has a potential broader application and will contribute well to the literature on stored product pest infestations.

Line 9: “introduced into the diets of human populations…

Line 48: “into the diets of several human populations

Line 76: “Stock rearing of Cadra cautella, Corcyra cephalonica, Plodia interpunctella, and Idaea inquinata has been maintained for 10 years”.  Has there been the addition of insects from the wild into these colonies over the past 10 years? 

Lines 111-132:  Species names need to be italicized.

Line 151: Table 2 “Nutritional characterization of the artificial diets of Idaea inquinata and of the Pyralid moths, as well as of the Moringa oleifera leaves, expressed as dry matter.” This table is confusing.  You may want two separate tables 1) the nutritional information of the artificial diet, 2) nutritional information of Moringa oleifera leaves

Author Response

We answered to the reviewer in the attached file

Reviewer 2 Report

The objective of this study was to verify the susceptibility of dried and powdered leaves of M. oleifera to stored product and a field pest species of moths. Tests were also carried out by using an artificial diet of which one-fourth consisted of dried and powdered leaves when rearing in the laboratory in order to determine if the effects of Moringa leaves on the development of these species were linked to their nutritional composition or to the presence of secondary components. This is a well presented manuscript.

General comments:

  • An important aspect in testing the survival of the pest on of oleifera leaves is the humidity of the product. We have no information on the moisture content of the product neither the equivalent equilibrium relative humidity that corresponds to dried product. This aspect is crucial since eggs were released into the culture media. Although storage insect's larvae can reproduce even at very low humidity, the adults are susceptible to low humidity and their lack of reproduction might be affected. Since the powdered leaves were dried, it is useful to know what the moisture content of dry and powdered leaves was.
  • We do not know which stage of the insect's development was affected by the dried leaves. Results were based on the emergence of adults from 100 eggs reared in different culture media.
  • Table 1 should contain basic information related to the experiment. The fact that each test was result of development from 100 eggs is missing. The temperature and relative humidity of the test is missing.

Specific comments:

Line 111, Cadra cautella, Corcyra cephalonica, Plodia interpunctella should be italic.

Line 112, Moringa oleifera should be italic.

Line 113, Idaea inquinata, M. oleifera should be italic.

Line 115, C. cautella, should be italic.

Line 126, I. inquinata, should be italic.

Lines 144 through 150 should be separated from Table 1 and should be part of the text format indented to left.

Lines 154 through 100 should be indented to left.

Author Response

(The authors gave the same response as above.)

Reviewer 3 Report

Interesting study with good data. It is one of the few studies that leaves are examined for their susceptibility to stored product insects, and not the opposite.

The flow of the introduction is good, but the authors can give more examples on stored product insects. The aims of the study can be expanded towards their novelty.

  1. Need to add authorities in full in the first time that a scientific name appears on the text.
  2. Provide one sentence with the materials (equipment) used.
  3. What was the normalization test used here? (Levene?). Also, did you make any transformation to meet the assumptions?
  4. Again (check also introduction and elsewhere) scientific names should be mentioned in full once, when they firstly appear on the text, and then the authors can use the abbreviated form, e.g. E. cautella etc. Scientific names should be in italics.
  5. ns is not enough. What was the actual value of P? The same holds in the case of the other areas where ns appears.

Table 1. My recommendation is to divide this into figures, in order to make it more comprehensive for the reader. Also, this approach is better as this will expand the length of the paper, given that as it stands now, it is rather short, and can be considered as a short communication.

Table 2. The 11.1 should have SE.

The Discussion is rather short. What is missing here is other similar works on infestations in dried botanicals (e.g. cigarette beetle etc.). Add another paragraph of practical recommendations at the end.

Author Response

(The authors gave the same response as above.)
